# Clinical Significance of Glycolytic Metabolic Activity in Hepatocellular Carcinoma

**DOI:** 10.3390/cancers15010186

**Published:** 2022-12-28

**Authors:** Joann Jung, Sowon Park, Yeonwoo Jang, Sung-Hwan Lee, Yun Seong Jeong, Sun Young Yim, Ju-Seog Lee

**Affiliations:** 1Department of Systems Biology, The University of Texas MD Anderson Cancer Center, Houston, TX 77030, USA; 2Division of Hepatobiliary and Pancreatic Surgery, Department of Surgery, Yonsei University College of Medicine, Yonsei 03722, Republic of Korea; 3Division of Hepatobiliary and Pancreas, Department of Surgery, CHA Bundang Medical Center, CHA University, Seongnam 46371, Republic of Korea; 4Department of Internal Medicine, Korea University College of Medicine, Seoul 02841, Republic of Korea

**Keywords:** liver cancer, hepatocellular carcinoma, cancer metabolism, glycolysis, transcriptome, survival, stem cells, immunotherapy, Tregs

## Abstract

**Simple Summary:**

Hepatocellular carcinoma (HCC) is among the most common cancers and causes about 830,000 deaths annually in the world. Metabolic reprogramming is a critical hallmark of HCC, enabling HCC cells to adapt to the high energy demands necessary for fast growth. However, the clinical relevance of metabolic alteration in HCC has not been systematically assessed. By performing cross-species comparison of genomic data from mouse and human tissues, we identified three distinct metabolic subtypes of HCC and uncovered clinical and molecular characteristics associated with three subtypes. Importantly, we showed that the high metabolic subtype is less susceptible to immunotherapy and uncovered a potential mechanism associated with resistance to immunotherapy.

**Abstract:**

High metabolic activity is a hallmark of cancers, including hepatocellular carcinoma (HCC). However, the molecular features of HCC with high metabolic activity contributing to clinical outcomes and the therapeutic implications of these characteristics are poorly understood. We aimed to define the features of HCC with high metabolic activity and uncover its association with response to current therapies. By integrating gene expression data from mouse liver tissues and tumor tissues from HCC patients (*n* = 1038), we uncovered three metabolically distinct HCC subtypes that differ in clinical outcomes and underlying molecular biology. The high metabolic subtype is characterized by poor survival, the strongest stem cell signature, high genomic instability, activation of EPCAM and SALL4, and low potential for benefitting from immunotherapy. Interestingly, immune cell analysis showed that regulatory T cells (Tregs) are highly enriched in high metabolic HCC tumors, suggesting that high metabolic activity of cancer cells may trigger activation or infiltration of Tregs, leading to cancer cells’ evasion of anti-cancer immune cells. In summary, we identified clinically and metabolically distinct subtypes of HCC, potential biomarkers associated with these subtypes, and a potential mechanism of metabolism-mediated immune evasion by HCC cells.

## 1. Introduction

Hepatocellular carcinoma (HCC) is among the most common cancers worldwide and causes about 830,000 deaths annually [1]. The incidence of HCC in the United States has increased over the past 25 years, to an estimated 41,260 new cases in 2022 [2]. Less than one-third of HCC patients are eligible for potentially curative treatments [3,4,5,6,7]; the vast majority of HCC patients present with advanced disease not amenable to curative treatments. Current standard first-line treatments for advanced HCC include targeted therapy with kinase inhibitors such as sorafenib and lenvatinib, which have antiangiogenic and antiproliferative effects, and immunotherapy with atezolizumab combined with bevacizumab [8,9,10,11]. However, kinase inhibitors appear to prolong HCC patients’ survival by only a few months, and immunotherapy only benefits patients who have HCC with viral etiologies [12]. Thus, there is a clear need to enhance our insight into the molecular development of HCC, which could lead to the discovery of new targeted therapies for HCC and/or effective strategies to extend the survival of HCC patients. 

Metabolic reprogramming is a critical hallmark of cancer [13,14], that enables cancer cells to adapt to the high energy demands necessary for fast growth. Indeed, many cancer cells acquire deregulated high metabolic activity that enables them to prosper even in a resource-limited microenvironment [15]. The best example is the surge in consuming glucose through anaerobic glycolysis, even in the presence of oxygen [16]. Considering that the liver is the primary site of metabolism in the body, it is not surprising to see highly dysregulated metabolism in HCC cells compared to that of normal hepatocytes [17,18]. However, the clinical relevance of metabolic alteration, particularly in glycolytic pathway, in HCC tumors has not been systematically assessed and clearly demonstrated. 

In a previous study [19], we showed that genomic signatures from mouse models are similar to those from human tumors and developed the approach known as “comparative systems genomics” that performs cross-species comparison of genomic data from mouse and human tissues to classify patients according to defined conditions from preclinical mouse models. In the current study, we adopted this method to uncover the clinical significance of metabolic alteration in HCC. 

## 2. Materials and Methods

### 2.1. Gene Expression Profile Data from Mouse Liver Tissues 

Gene expression profile data from mouse liver tissues were generated as described previously [20]. Eight week-old C57B/l6 male mice were fed a regular diet (ad libitum) with or without 20% glucose or fructose for 24 h in drinking tap water (*n* = 6 per group and 18 in total). Mice were euthanized for collection of RNA at 14 h after the start of light period in the animal housing unit. Total RNA was extracted from liver tissues of mice and used to generate gene expression data via the Agilent microarray platform (SurePrint G3 Mouse GE v2 8x60K Microarray). Data are available in the National Center for Biotechnology Information’s Gene Expression Omnibus (GEO) database (GSE92502).

### 2.2. Gene Expression and Clinical Data from Human HCC 

Gene expression data and clinical data were described in earlier studies [21,22,23,24,25,26]. Briefly, gene expression data from the Fudan cohort were obtained from the GEO database (accession number GSE14520) [21]. Gene expression data from the Korean cohort were generated using the Illumina microarray platform human-6 v2 and v4 (accession numbers GSE16757, GSE43619) [22,23]. Gene expression data from the Modena cohort were obtained from GEO databases (accession number GSE54236) [24]. Gene expression data from Zhongshan hospital cohort were obtained from National Omics Data Encyclopedia (NODE, accession number OEP000321) [25]. We also included gene expression data from The Cancer Genome Atlas (TCGA) HCC project in this analysis [26]. Appendix A shows the summary of data sets in all five cohorts. All patients had undergone surgical resection as their primary treatment for HCC.

### 2.3. Identification of Hepatic Glycolytic Gene Expression Signature from Mouse Liver

To identify genes reflecting high glycolytic activity in mouse liver tissue, we first selected genes whose expression is significantly induced by fructose-feeding or glucose-feeding in mouse livers, yielding 1960 genes for fructose-specific induction and 2022 genes for glucose-specific induction. By comparing the two gene lists, 416 overlapping genes were identified as glycolytic genes, and their expression patterns were considered to be the hepatic glycolytic gene expression signature (Figure 1). Later, identified gene sets were subjected to ingenuity pathway analysis (September release 2022), which revealed a myriad of affected signaling pathways and functional categories.

### 2.4. Data Analysis

Collected gene expression data were transformed and normalized as described previously [20]. BRB-ArrayTools, v4.6, a freeware program from the National Cancer Institute (https://brb.nci.nih.gov/BRB-ArrayTools/ accessed on 11 June 2022), was used for analyzing the data and building a predictive model [27]. Cluster (v 3.0) and TreeView (v 1.6) were used to generate a heatmap of gene expression data [28]. R language (http://www.r-project.org, v 4.1.1 accessed on 15 September 2021) was used for statistical analysis. Somatic copy number alterations in TCGA data were determined by profiling HCC on Affymetrix SNP 6.0 arrays and analysis by GISTIC 2.0 [29].

Before pooling mouse and human gene expression data for performing cross-species analysis, expression data of orthologous genes in both data sets were independently converted to z-scores (z = (x − mean)/standard deviation) [19]. A Bayesian compound covariate prediction (BCCP) algorithm was used to estimate the probability that a particular human HCC sample would have a given gene expression pattern from mouse tissue [19,30]. Gene expression data from mouse tissue (training sets) were combined to form a predictor according to a BCCP model. The robustness of the predictor was estimated by a misclassification rate determined using leave-one-out cross-validation during training. Sensitivity and specificity of predicting sugar-fed liver tissue in the mouse training set were 1.0 and 1.0, respectively. The BCCP model estimated the probability that an individual human HCC sample would have high or low glycolytic activity and trichotomized tumors according to Bayesian probability (cutoff of 0.8 and 0.2). 

To generate the hepatic stem cell (HSC) probability of HCC tumors, we applied a previously established HSC signature to gene expression data from HCC tumors as described previously [31,32]. 

### 2.5. Gene Expression Data from HCC PDX Models

HCC PDX tumors were established by Crown Bioscience as described earlier [33,34]. mRNA expression data from PDX tumors were generated by Illumina HiSeq2500 platform. For bioinformatics analysis of transcriptome sequencing data, RNAseq raw data were first cleaned up by removing contamination mouse mRNA reads that preferentially mapped to mouse genome (UCSC MM10). Clean reads were mapped to reference genes (ENSEMBL GRCh37.66) by Bowtie (v 1.2.3), and gene expression was calculated by MMSEQ (v 1.0.10). The hepatic glycolytic gene expression signature was applied to gene expression data from PDX model to stratify them to 3 subtypes. 

## 3. Results 

### 3.1. Gene Expression Signature Reflecting Glycolytic Activity from Mouse Liver Tissue 

We examined the glycolytic activity of HCC tumors by using a comparative cross-species genomic approach that integrates genomic data from the well-defined experimental conditions of animal models into those from human HCC. To do this, genes whose expression is significantly correlated with glycolytic activity in mouse liver were identified by applying a Student’s t-test to gene expression profile data from liver tissues of mice fed with fructose or glucose versus control tap water. Overlapping expression of 416 genes was identified as a hepatic glycolytic signature (*p* < 0.01, Figure 1, and Appendix A). As expected, the upregulated genes included metabolic genes such as *Psat1*, *Fut1*, *Gpi1*, *Rpia*, *Acaca*, and *Pklr*, suggesting that the signature well reflect high metabolism in the liver. Hereafter, we refer to the defined signature as the glycolysis metabolic (GM) signature.

To further reveal the underlying biological activity of the GM signature in the liver, we next performed gene network analysis of the GM signature by applying Ingenuity Pathway Analysis. Not surprisingly, it revealed the glycolysis pathway as one of the most activated pathways in the GM signature (Appendix A). Other activated pathways included the mTOR pathway, reflecting high energy consumption, and the cell growth pathway, suggesting that highly glycolytic activity leads to high cellular energy production and cell growth. Interestingly, the ferroptosis signaling pathway was also activated by high glycolysis, suggesting that high metabolic activity may increase oxidative stress, which is the foundation of ferroptotic cell death [35,36]. In agreement with this, the NRF2-mediated oxidative stress response pathway was also activated by high glycolysis.

### 3.2. Association of Hepatic Metabolic Activity with Prognosis of Patients with HCC

Having generated a gene expression signature reflecting high metabolic activity in liver, we next tested the clinical relevance of hepatic glycolytic activity in primary HCC tumors from patients by extracting the expression of GM signature genes from patients’ tumors and comparing them with the GM signature from mouse liver tissues. To validate clinical association of GM signature in HCC, we built a stratifying prediction model with the mouse GM signature and directly applied it to the genomic data from HCC tumors. Expression data from the mouse GM signature (training set) were used to generate a BCCP that estimated the probability of high metabolic activity in each HCC tumor (Figure 2A). Patients in the Fudan HCC cohort (*n* = 242) were trichotomized according to Bayesian probability (<0.2, 0.2 to 0.8, >0.8), which classified 69, 108, and 65 patients into high, middle, and low metabolic activity subgroups, respectively (Figure 2B). Kaplan–Meier plots for overall survival (OS) of patients in the Fudan cohort showed significant differences in OS after treatment (*p* = 1.6 × 10^−6^ by log-rank test) among the three subgroups (Figure 2C), strongly indicating that high glycolytic activity in HCC significantly contributes to patients’ clinical outcome after treatment.

We next examined the correlation of glycolytic activity with patients’ prognosis in four additional HCC cohorts (ZhongShan cohort, *n* = 159; TCGA cohort, *n* = 371; Korean cohort, *n* = 188; and Modena cohort, *n* = 78, Figure 2A and Appendix A). When the BCCP used in the Fudan cohort was applied to the four additional cohorts, Kaplan–Meier plots of all cohorts showed significant differences in OS among the three GM subtypes (*p* = 1.0 × 10^−5^ for ZhongShan, *p* = 0.005 for TCGA, *p* = 0.02 for Korean, and *p* = 0.002 for Modena by log-rank test, Figure 3). Together, the results from all five cohorts (*n* = 1038) clearly demonstrated a strong association between the high glycolytic activity and poor OS rates in HCC.

### 3.3. Prognostic Significance of GM Subtypes

To quantify the prognostic weight of glycolytic activity in combination with other critical clinical features, we performed univariate Cox proportional analyses with clinical features from the Zhongshan cohort, which had the most complete set of clinical data. In addition to tumor size and Barcelona Clinics Liver Cancer (BCLC) stage, which are well-known variables associated with OS, the GM signature was a statistically significant predictor of OS (Table 1). In multivariate analysis with analyzed variables together, the high GM subtype was independent prognostic predictor for OS as evidenced by high hazard ratio of 2.97 (95% confidence interval, 1.72−5.12 and *p* = 8.5 × 10^−5^).

We next estimated how GM subtypes are independent across the standard clinical stages. When the GM signature was applied to patients with BCLC stage A, which is considered early stage HCC [37], patients with the high GM subtype had worse OS outcomes than patients with the middle and low GM subtypes (Appendix A). Taken together with Cox analysis, this observation suggests that GM signature retains its prognostic significance even after the classic clinicopathological variables have been taken into account. 

### 3.4. Mutations and Genomic Alterations Associated with GM Subtypes 

We next assessed the association of genomic characteristics with GM subtypes in the TCGA cohort, for which genomic data were available. to gain additional insight into each subtype’s biology. We found no differences in mutation burden among the three GM subtypes (Figure 4A). However, alterations of genomic copy number differed significantly among the subtypes, with the high GM subtype having the most (Figure 4B). We next sought to identify somatic mutations significantly associated with the subtypes (Appendix A). *TP53* mutations were associated with the high and middle GM subtypes (Figure 4C). *FAM47A* mutations were associated with the high GM subtype, and *CTNNB1* (encoding β-catenin) mutations were associated with the low GM subtype. *ALB* (coding albumin) mutations were significantly less frequent in the high GM subtype, suggesting a potential connection of loss of albumin activity in regulation of the glycolytic pathway. 

### 3.5. Potential Sensitivity to Immunotherapy among GM Subtypes

The combination of bevacizumab, which targets VEGF, and atezolizumab, an immune checkpoint inhibitor that selectively targets PD-L1, has yielded encouraging results in HCC patients [11]. Therefore, we estimated each GM subtype’s potential response to immunotherapy using tumor immune dysfunction and exclusion (TIDE) scores, which reflect resistance to immune checkpoint inhibitors [38]. Interestingly, most tumors (86.8%) in the high GM subtype showed high TIDE scores (Figure 5A), suggesting that HCC patients with high metabolic activity would not have substantial benefit from immunotherapy. Moreover, the GM probability was positively correlated with TIDE score (r = 0.3381, *p* = 1.98 × 10^−11^) (Figure 5B), further supporting the association of the high GM subtype with resistance to immunotherapy. 

To uncover the biology underlying the low response of high metabolic HCC tumors to immunotherapy, we explored the percentage of immune cells in tumors by analyzing their gene expression data using the previously established CIBERSORT algorithm [39] (Figure 5C). Interestingly, the fraction of immunosuppressive regulatory T cells (Tregs) was significantly higher in the high GM subtype (Figure 5D), suggesting that high metabolic activity in the tumor microenvironment may trigger activation of Tregs, leading to the low response to immunotherapy in high GM HCC tumors. Furthermore, the fraction of naïve M0 macrophages was also higher in high GM HCC tumors (Figure 5E), suggesting that that absence of active anti-cancer macrophages may contribute to the poor response of high GM tumors to immunotherapy. In agreement with these observations, the estimated level of myeloid-derived suppressor cells from the TIDE analysis was significantly higher in high GM subtype than in the other subtypes (Appendix A). Similarly, expression of major inhibitors of immune checkpoints CTLA-4 and PD-1 were significantly higher in the high GM subtype (Appendix A), further supporting the notion that high metabolic activity may suppress immune activity. 

### 3.6. Stem Cell Characteristics in GM Subtypes 

We next sought to correlate the GM subtypes with stem cell characteristics by applying a previously established HSC signature to gene expression data from HCC tumors [31]. The high GM subtype showed significantly higher HSC probability than the middle and low GM subtypes (*p* < 0.001 by *t* test, Figure 6A), suggesting that high metabolic activity in HCC might be driven by genetic or genomic switches activated in HSCs. Consistently, HSC probability showed significant correlation with GM probability (r = 0.6269, *p* = 4.9 × 10^−49^, Figure 6B), further supporting a close relationship between high metabolic activity and HSC features in HCC. We next examined the expression of cancer stem cell markers. Not surprisingly, expression of many stem cell markers were significantly higher in the GM high subtype than in the other GM subtypes (Figure 6C). In particular, expression of well-known hepatic stem markers such as *AFP*, *KRT19*, and *EPCAM* was significantly higher in high GM subtypes (Figure 6D), and SALL4 was the most significantly correlated transcription regulator with the high GM subtype. 

### 3.7. GM Subtypes in Preclinical Models

We next examined whether the GM subtypes’ metabolic characteristics are preserved in preclinical models of HCC. We applied the BCCP GM predictor to the genomic data from 77 HCC patient-derived xenograft (PDX) tumors. GM gene expression patterns were well conserved in these tumors (Appendix A), indicating that metabolic characteristics are well preserved in PDX tumors. Established PDX models appeared to be stable, as illustrated by the fact that there was no significant difference in number of passages in PDX models among subtypes (Appendix A), suggesting that metabolic features in primary tumors do not fade out over passages. 

## 4. Discussion 

In the current study, by integrating gene expression profile data from human HCC tumors with those from mouse models, we identified three metabolically distinct HCC subtypes that are significantly different in prognosis and potential response to standard treatment with immunotherapy. Analysis of genomic data from multiple sources uncovered connections between high metabolic activity and several pathways that might account for poor prognosis in HCC patients and identified potential prognostic markers. Our results may lead to new opportunities in advancing molecular classification of HCC patients and providing potential treatment guidance. 

To develop the gene expression signature reflecting hepatic metabolic activity and prognosis, we used a supervised approach combined with validation in multiple cohorts of HCC patients. This approach yielded several lines of evidence that support significant association of metabolic activity with prognosis in HCC. First, its strong association with prognosis was tested and validated in five independent HCC cohorts. Second, the GM signature could identify patients at high risk of shorter OS among those with early stage HCC (BCLC A stage). Last, in multivariable Cox regression analysis, the GM signature was one of the most significant predictors of OS. 

In our study, HCC tumors with high metabolic activity were characterized by high genomic instability, as reflected in their numerous copy number alterations and high frequency of *TP53* mutation. This is in agreement with previous reports showing poor prognostic features of HCC with *TP53* mutations [40]. Interestingly, the TIDE score, which reflects potential response to immunotherapy, showed that the high GM subtype would be the least responsive to immunotherapy. The strong connection of high genome instability with poor response to immunotherapy in the high GM subtype is consistent with previous studies showing that chromosome instability is significantly associated with immune evasion and with poor response to immunotherapy [41,42]. The predicted poor response to immunotherapy of the high GM subtype is further supported by a high expression of immune checkpoint regulatory genes, such as those encoding CTLA-4 and PD-1, in that subtype. Our observation of poor response of high metabolic tumors to immunotherapy is further supported by clinical analysis of ^18^F-FDG PET/CT imaging data to assess the response of patients with metastatic melanoma to immunotherapy [43]. A meta-analysis of 24 published reports showed that tumors’ high metabolic activity, reflected in baseline metabolic tumor volume, maximum standardized uptake value, and total lesion glycolysis, was significantly associated with poorer OS of patients after immunotherapy. More interestingly, we found that the estimated Treg cell fraction in the tumor microenvironment was highest in high GM subtype, indicating that poor response of high metabolic HCC tumors to immunotherapy might result from the activation of Tregs that suppress anti-cancer immune activity [44]. High glycolytic activity of HCC tumors eventually leads to accumulation of the glycolysis by-product lactic acid in the tumor microenvironment [45]. A recent study showed that Tregs can effectively use lactic acid as metabolic fuel for proliferation [46], suggesting that lactic acid might account for the aggregation of Tregs in HCC tumors’ high metabolic microenvironment.

The high GM subtype is also characterized by strong stem cell features, as reflected in their high stemness scores and high expression of HSC markers such as *AFP*, *KRT19*, *EPCAM*, and *SALL4*. While the gene expression patterns of high GM HCC tumors were substantially similar to fetal HSCs, it is currently unknown whether this high similarity reflects the origin of cancer cells or the high fraction of cancer stem cells in the tumor mass. Invasion is a common event in poor-prognosis tumors with stem cell features [47]. Since zinc-finger transcription factor SALL4 is a stem cell factor triggering invasion and migration of cancer cells [48], it might contribute to a metastatic phenotype in high metabolic HCC tumors. Interestingly, recent studies showed that SALL4 is a neosubstrate of the molecular glue thalidomide and its derivatives that degrade its target proteins via the E3 ligase complex system [49,50], suggesting that thalidomide and its derivatives could be used for treatment of high metabolic HCC tumors in the future. 

The current study was a genomic analysis with limited exploration of the biology of high metabolic activity in association with poor prognosis and poor response to immunotherapy. However, the GM signature had a solid association with clinical outcome in HCC patients. For validation of high metabolic activity’s association with resistance to immunotherapy in patients with HCC, more in vitro and in vivo study will be necessary. Nevertheless, the newly identified oncogenic pathways associated with metabolic activity will offer opportunities to identify novel therapeutic targets for HCC. Moreover, the GM signature is well conserved in PDX models, offering a tool for selecting the best preclinical models for future study.

## 5. Conclusions

In summary, our finding suggested that high glycolytic activity in HCC is significantly associated with poor survival of patients. In depth analysis of metabolism associated genomic traits further suggested that high glycolytic activity in HCC may trigger activation of cancer stem cells and evasion of cancer cells from immune surveillance. 

## Figures and Tables

**Figure 1 cancers-15-00186-f001:**
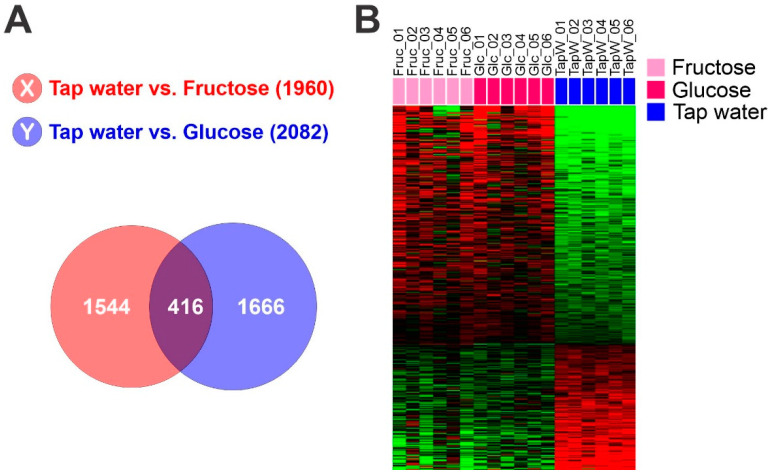
Hepatic glycolytic gene expression signature from mouse liver. (**A**) Venn diagram of genes selected by a two-sample *t* test. The red circle (gene list X) represents genes differentially expressed between liver tissues from mice fed with tap water and those fed with fructose water. The blue circle (gene list Y) represents genes differentially expressed between liver tissues of mice fed with tap water and those fed with glucose water. We applied a cut-off *p*-value of less than 0.01 to retain genes whose expression is significantly different between the two groups of tissues. (**B**) Expression patterns of selected genes in the Venn diagram. Gene expression data from livers of mice fed with fructose, glucose or control tap water were selected from 416 overlapping genes.

**Figure 2 cancers-15-00186-f002:**
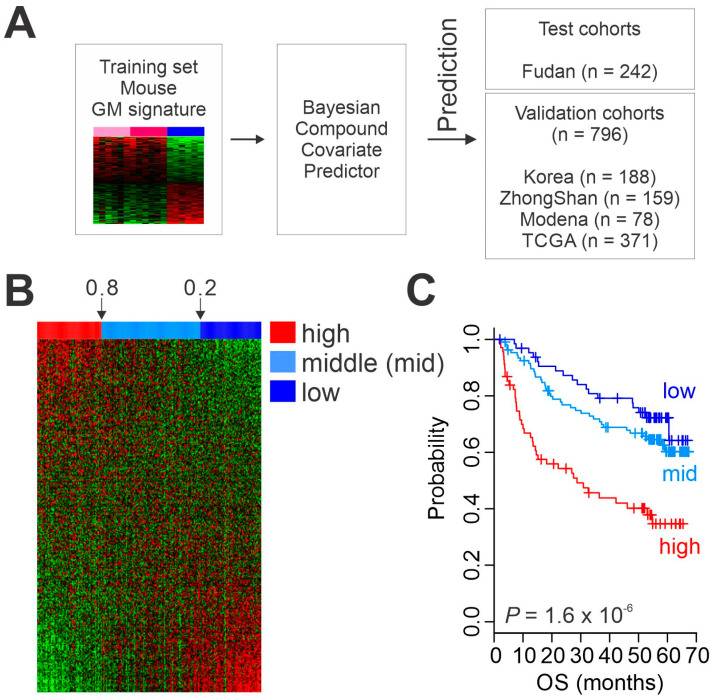
Clinical association of metabolic activity in hepatocellular carcinoma (HCC). (**A**) Schematic diagram of the prediction model. (**B**) Heatmap of glycolysis metabolic (GM) gene expression signature in patients from the Fudan cohort. (**C**) Kaplan–Meier plots of overall survival (OS) of HCC patients in the Fudan cohort stratified by GM subtype. TCGA, The Cancer Genome Atlas.

**Figure 3 cancers-15-00186-f003:**
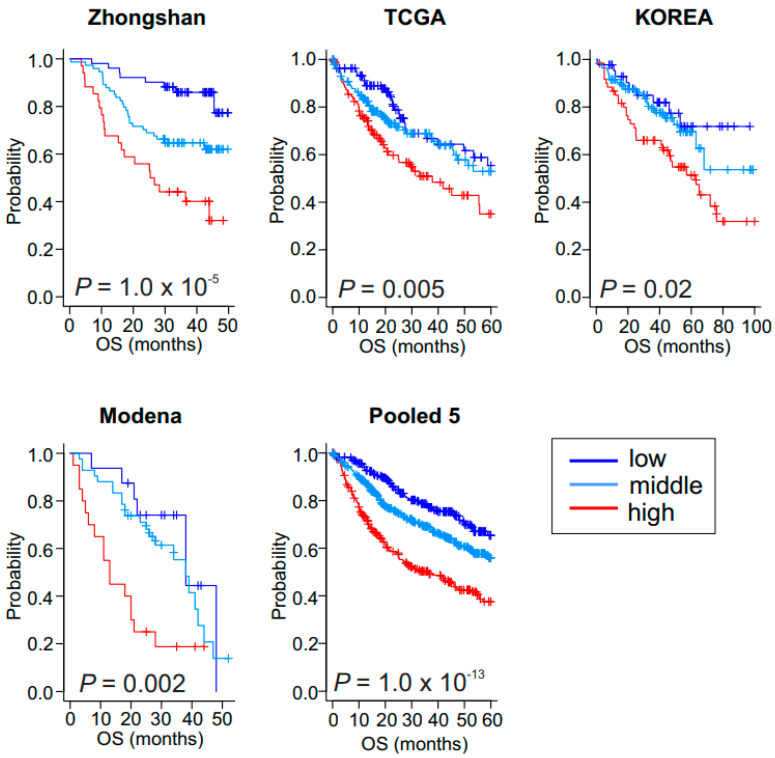
Validation of clinical association of low, middle, and high glycolysis metabolic subtypes in hepatocellular carcinoma. Kaplan–Meier plots of overall survival (OS) in patients in the validation cohorts. Zhongshan cohort (*n* = 159), Korea cohort (*n* = 188), The Cancer Genome Atlas (TCGA) cohort (*n* = 371), Modena cohort (*n* = 78), and pool of five cohorts (*n* = 1038).

**Figure 4 cancers-15-00186-f004:**
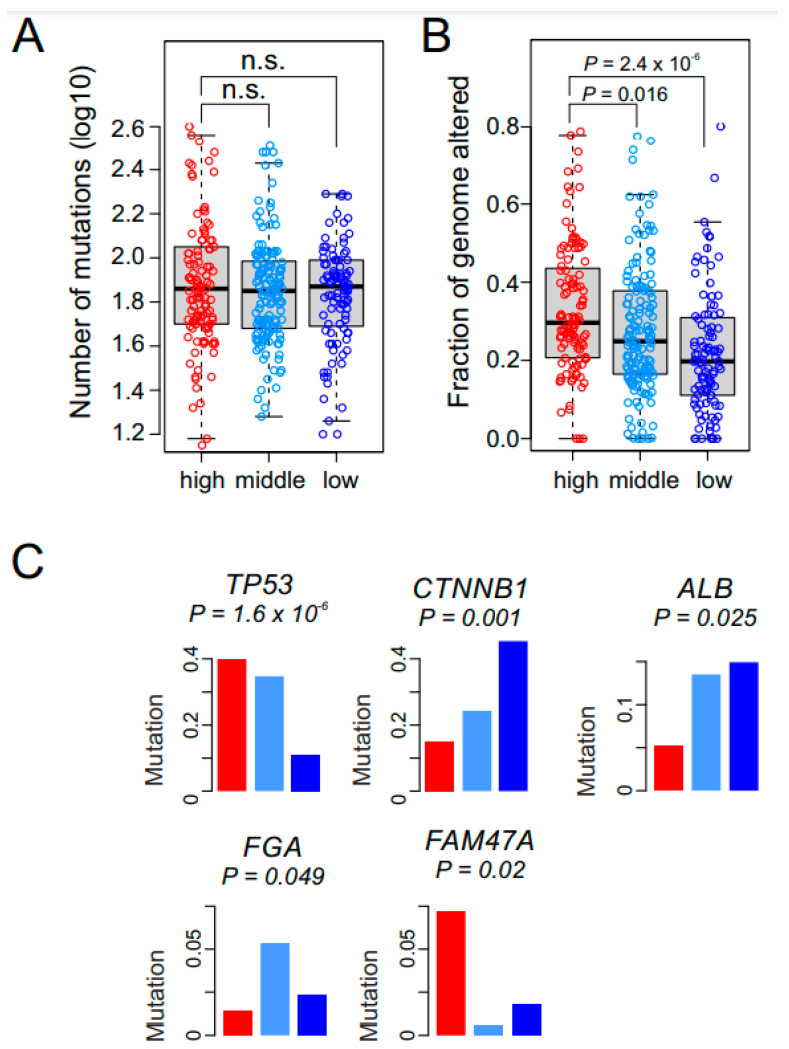
Genomic alterations in the glycolysis metabolic (GM) subtypes. (**A**) Bee swarm box plots for number of nonsynonymous mutations in GM subtypes (*n* = 367). No significant difference is observed among the GM subtypes. In the box plots, the boundary of the box indicates the 25th to 75th percentile, and the black line within the box marks the mean. Whiskers above and below the box indicate the 10th and 90th percentiles. Circles represent the number of mutations in each tumor. (**B**) The fraction of the genome altered by copy number gain and loss was estimated by GISTIC2 analysis in each tumor (*n* = 367). The high GM subtype has significantly higher alterations than the other two subtypes (all *p* < 0.05 by Student *t* test). (**C**) Somatic mutations associated with GM subtypes in TCGA cohort. Mutation rates of each gene are presented as fraction within subtypes. Red, light blue, and dark blue represent the high, middle, and low GM subtypes, respectively.

**Figure 5 cancers-15-00186-f005:**
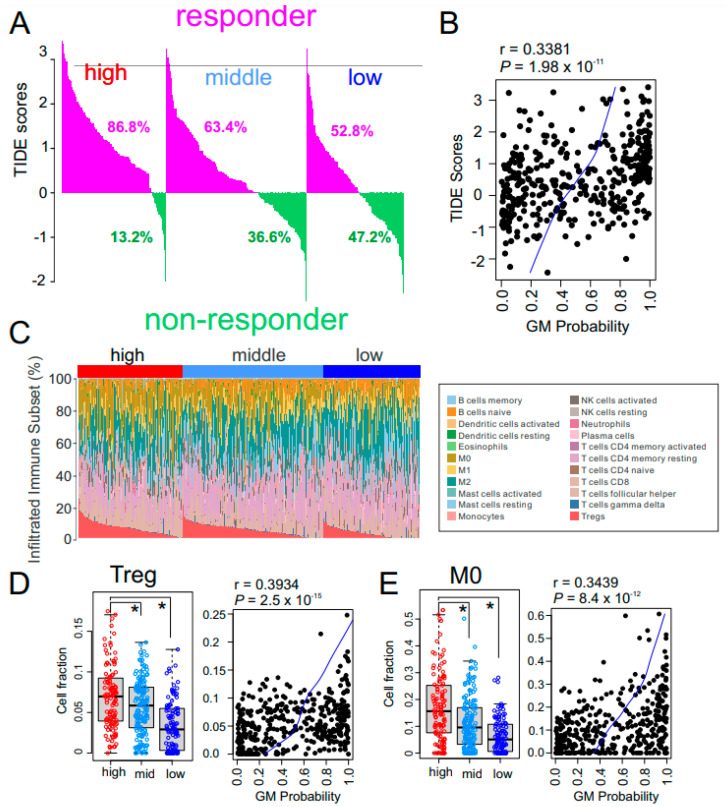
Immune characteristics of the glycolysis metabolic (GM) subtypes. (**A**) Waterfall plots for the response rates to immunotherapy predicted by the tumor immune dysfunction and exclusion (TIDE) algorithm in The Cancer Genome Atlas Liver Hepatocellular Carcinoma (TCGA-LIHC) cohort (*n* = 371). Numbers below waterfall plots represent the fraction of responders in the patients with each GM subtype. (**B**) Scatter plot for the correlation between TIDE score and GM probability in the TCGA cohort (*n* = 371). Blue line indicates locally weighted scatterplot smoothing (lowess) regression. (**C**) The pattern of infiltrations of 22 immune subsets according to GM subtype from fetal liver signatures predicted by the CIBERSORT algorithm in the TCGA cohort. (**D**,**E**) Box and scatter plots of fraction of regulatory T cells (Tregs) (**D**) and M0 macrophages (**E**) in GM subtypes. Relative fraction of each immune-subset was normalized by mean and standard deviation across the samples. In the scatter plots, blue line indicates locally weighted scatterplot smoothing (lowess) regression. In the box plots, the boundary of the box indicates the 25th to 75th percentile, and the black line within the box marks the mean. Whiskers above and below the box indicate the 10th and 90th percentiles. Each circle represents the fraction of indicated immune cells in each tumor. * *p* < 0.001 by Student *t* test.

**Figure 6 cancers-15-00186-f006:**
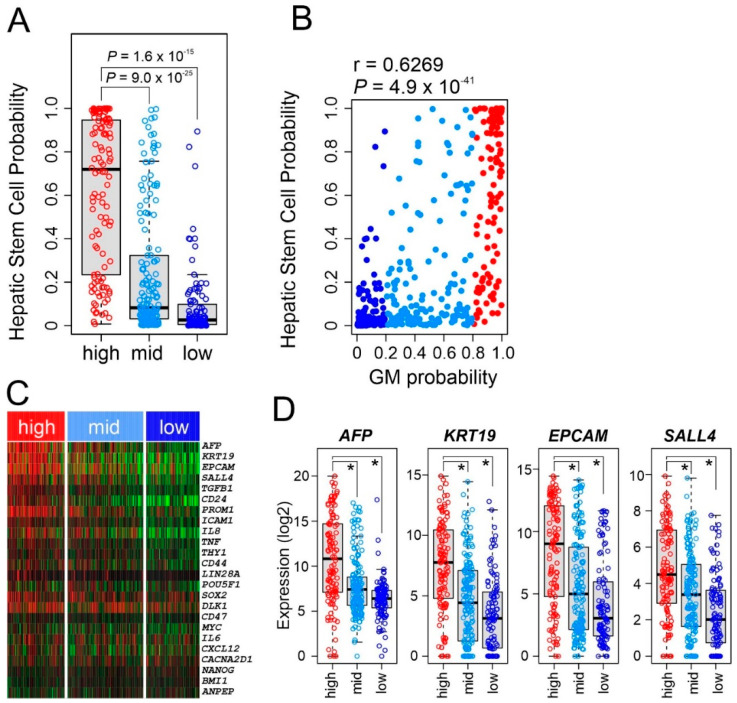
Stem cell characteristics of glycolysis metabolic (GM) subtypes. (**A**) Hepatic stem cell probability of hepatocellular carcinoma tumors in GM subtypes in The Cancer Genome Atlas (TCGA) cohort. In the box plots, the boundary of the box indicates the 25th to 75th percentile, and the black line within the box marks the mean. Whiskers above and below the box indicate the 10th and 90th percentiles. Each circle represents the fraction of the indicated immune cells in each tumor. Student *t* test. (**B**) Scatter plot for the correlation between hepatic stem cell probability and GM probability in the TCGA cohort (*n* = 371). (**C**) Heatmap for expression of major stem-cell markers according GM subtype in the TCGA cohort. (**D**) Box plots of expression of stem cell markers according to GM subtype in the TCGA cohort. * *p* < 0.001 by Student *t* test.

**Table 1 cancers-15-00186-t001:** Univariate and multivariate Cox regression analyses of overall survival in Zhongshan cohort.

Characteristic	Univariate		Multivariate	
	Hazard Ratio (95% CI)	*p* Value	Hazard Ratio (95% CI)	*p* Value
Patient sex (M or F)	0.75 (0.4–1.41)	0.381		
Age (>60 years or not)	0.8 (0.44–1.45)	0.47		
AFP (>300 ng/mL or not)	3.12 (1.83–5.34)	<0.001	2.75 (1.53–4.91)	<0.001
Cirrhosis (yes or no)	1.28 (0.69–2.35)	0.42		
Tumor size (>6 cm or not)	3.53 (1.97–6.32)	<0.001	5.26 (1.86–14.8)	0.001
BCLC stage (B/C/D or 0/A)	2.77 (1.51–5.09)	0.001	0.57 (0.23–1.4)	0.23
GM signature (high or mid/low)	2.97 (1.72–5.12)	<0.001	1.84 (1.04–3.25)	0.033

CI, confidence interval; AFP, alpha-fetoprotein; BCLC, Barcelona Clinic Liver Cancer; GM, glycolytic metabolism.

## Data Availability

All data used in current study are publicly available as shown in Appendix A.

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
