# Peer review of "Clinical Significance of Glycolytic Metabolic Activity in Hepatocellular Carcinoma"

_cancers, 2022, doi:10.3390/cancers15010186_

Round 1

Reviewer 1 Report

First of all, I would like to congratulate you for the article, the truth is that the field of glycolytic metabolic activity and cancer is a very interesting one. The elucidation of some aspects of the cellular metabolism of cancer cells could help to establish new lines of treatment that could be more effective and directed at the key aspects altered in tumour cells.

A study in humans is lacking. I understand that this type of study is more complex due to the stricter regulation, but since only tap water, fructose, and glucose are used, I think it might be easier to obtain a human adult population sample to compare with HCC cohort data existing and that you comment in the article.

In addition, I think that there are several aspects that, in my opinion, should be corrected before the article can be definitively published, and that I detail below:

· In the introduction, the bibliography should be reviewed a little more thoroughly, since some citations are from a long time ago and I think they could be more up-to-date. As an example in line 40, in which it is said that cases of liver cancer will double in the next 20 years and the quotes that are made are from the years 2004 and 1999.

· I did not seem to see the number of mice included in the study, and despite the fact that Figure 1B suggests that there could be 18, it should be specified in the material and methods section.

· In the Figure 1, the number of genes of subset Y is not seen in the Venn diagram, if there are 2082 in total, and 416 are common with subset X, I understand that the missing part in the diagram should put 1666.

· The link to the BRB website is not that one. By putting the article, it redirects to the following: https://brb.nci.nih.gov/BRB-ArrayTools/. Is that link the right one?

· Lines 108 and 109 refer to the use of software used for data analysis, but the version is not mentioned. The same happens on lines 131 and 132 with Bowtie and MMSEQ.

· No mention is made in the methodology of the Ingenuity Pathway Analysis application, but mention is made of its use in lines 149-150 for the gene network analysis of the GM signature.

· An explanation of the univariate and multivariate Cox analysis carried out is necessary, since the data are exposed, but its conclusions are not explained and why this type of survival analysis is carried out.

· Missing reference to figure 4C (paragraph between lines 221 and 225?).

· On line 243, in the legend of figure 4, reference is made to GISTIC2, which is also not referred to in the methodology.

· Line 271 should be figure 5E and not figure 4E.

· In the paragraph on lines 280-292 reference is made to figures that do not appear in the article or in the supplementary material.

· The bibliography has many citations to articles older than 10 years, and there are also many citations to articles by the authors themselves. Mention should be made of more articles from other groups that work on the same or similar topics.

Reviewer 2 Report

Jung et. al., in their current study integrated the gene expression profiling data from human HCC tumors with those of mice using the comparative system genomics approach. In their current study, they identified three metabolically distinct HCC subtypes that were different in respect of their detection and response to the treatment therapies. The authors highlighted the connection between high metabolic activity and several underlying pathways that could be crucial in identifying potential markers and targets, to develop new therapies. The strongest part of the review is that the genomic data was collected from multiple sources, combined with validation in multiple cohorts of HCC patients. The outcomes underlined in the study are very important and would be useful for the researchers working in the field of HCC.

Minor comments:

On page 2, line 54-55, glucose and aerobic glycolysis are with hyperlink and in different font. Not sure if that was intentional or by mistake. If it was intentional, I feel there is no need for the hyperlink, since the readers are already aware of this basic concept, or if needed they can go to the reference.

The sex of the mice used for the gene expression profiling is missing

On page 2, line 88, is “Identification of hepatic glycolytic gene expression signature from mouse liver” the heading? Authors are advised to do proper formatting if its heading or rephrase the sentence if it’s not a heading.

On page 3, line 109, the font of the link needs to be changed.

How the expression data of orthologous genes in both human and mice data sets were standardized independently is not clear?

On page 4, line 126 heading needs formatting.

On page 4, lines 138-146, the text needs to be formatted.

On page 4, line 158 has extra spacing

Authors are advised to check the whole manuscripts for the spelling errors, formatting and grammar.

Author Response

Jung et. al., in their current study integrated the gene expression profiling data from human HCC tumors with those of mice using the comparative system genomics approach. In their current study, they identified three metabolically distinct HCC subtypes that were different in respect of their detection and response to the treatment therapies. The authors highlighted the connection between high metabolic activity and several underlying pathways that could be crucial in identifying potential markers and targets, to develop new therapies. The strongest part of the review is that the genomic data was collected from multiple sources, combined with validation in multiple cohorts of HCC patients. The outcomes underlined in the study are very important and would be useful for the researchers working in the field of HCC.

Minor comments:

On page 2, line 54-55, glucose and aerobic glycolysis are with hyperlink and in different font. Not sure if that was intentional or by mistake. If it was intentional, I feel there is no need for the hyperlink, since the readers are already aware of this basic concept, or if needed they can go to the reference.

> Because we submitted manuscript without using publisher’s template, our manuscript was converted to publisher’s format by editorial office later. Some errors were generated during conversion. We will correct all of errors occurred during conversion of format in revised manuscript.  

The sex of the mice used for the gene expression profiling is missing

> all mice were male and relevant information is updated in revised manuscript.

On page 2, line 88, is “Identification of hepatic glycolytic gene expression signature from mouse liver” the heading? Authors are advised to do proper formatting if its heading or rephrase the sentence if it’s not a heading.

> Yes, it is heading for subsection and corrected in revised manuscript.

On page 3, line 109, the font of the link needs to be changed.

> It is corrected in revised manuscript.

How the expression data of orthologous genes in both human and mice data sets were standardized independently is not clear?

> For comparison of two data sets, we converted them to Z-scores via subtracting them with mean and later dividing them by standard deviation, z=(x-mean)/std. This information is updated in revised manuscript.  

On page 4, line 126 heading needs formatting.

> corrected.

On page 4, lines 138-146, the text needs to be formatted.

> corrected.

On page 4, line 158 has extra spacing

> corrected.

Authors are advised to check the whole manuscripts for the spelling errors, formatting and grammar.

> Manuscript will be edited by professional editor at MD Anderson Cancer Center before publication.

Reviewer 3 Report

Lee et al. focused on the role of glycolytic metabolic activity in hepatocellular carcinoma. Put aside the results first; some of the basic information provided in this manuscript could be considered the main reasons to reject it. The reasons are listed below.

First, the novelty of this study is poor. The roles of glycolytic metabolic regulators in HCC were reported several times before, such as in PMID: 32576292, PMID: 32631382, PMID: 32407813, PMID: 32479426, and so on.

Second, the hepatic glycolytic gene identification strategy was wrong. The authors considered the differential expression of genes between liver cells before and after feeding glucose or fructose to be a glycolytic gene based on the material and methods. In general, after eating sugar, the first active physiological process in the liver is glycogen synthesis. Besides, the DEGs between the water and fructose groups more accurately represent the fructose-glucose conversion pathway. Moreover, in healthy mice, the hepatocytes mainly carry out aerobic respiration rather than glycolysis. In summary, the hepatic glycolytic gene identification strategy was illogical, and the so-called "hepatic glycolytic genes" were not glycolytic genes.

Third, some methods were too simple to understand the specific processes. For example, detailed methods of PDX tumor generation were absent. Besides, detailed methods related to gene expression data processing were not included. The self-citation rate is too high.

Furthermore, some minor suggestions and questions are listed below.

  • The format was chaotic. Particularly lines 54, 88, and 136-147.
  • Why was the PDX tumor sequencing data mapped to UCSC MM10? PDX tumors indeed originate from human tissue.
  • Why were the DEGs analyses conducted using a t-test? Limma or other DEGs' calling tools would be a better choice.
  • Why not filter DEGs based on the adjusted p-value?
  • Why were clinical traits, which show no significance in univariate Cox regression analysis, embraced in multivariate Cox regression analysis?
  • There was no figure 6 in this manuscript.

Author Response

Response to reviewer is attached 

Round 2

Reviewer 3 Report

The authors responded to most of my concerns. I have no further questions.